# Phylogenetic Analysis of Russian Native Sheep Breeds Based on mtDNA Sequences

**DOI:** 10.3390/genes14091701

**Published:** 2023-08-27

**Authors:** Olga Koshkina, Tatiana Deniskova, Arsen Dotsev, Elisabeth Kunz, Marina Selionova, Ivica Medugorac, Natalia Zinovieva

**Affiliations:** 1L.K. Ernst Federal Research Center for Animal Husbandry, Dubrovitsy 60, Podolsk Municipal District, Moscow 142132, Russia; olechka1808@list.ru (O.K.); asnd@mail.ru (A.D.); n_zinovieva@mail.ru (N.Z.); 2Population Genomics Group, Department of Veterinary Sciences, Ludwig-Maximilians-University Munich, 82152 Munich, Germany; elisabeth.kunz@gen.vetmed.uni-muenchen.de (E.K.); ivica.medjugorac@gen.vetmed.uni-muenchen.de (I.M.); 3Timiryazev Agricultural Academy, Russian State Agrarian University-Moscow, Timiryazevskaya Street, 41, Moscow 127550, Russia; m_selin@mail.ru

**Keywords:** local sheep breeds, mtDNA, origin, phylogeny

## Abstract

Eurasia is represented by all climatic zones and various environments. A unique breed variety of farm animals has been developed in Russia, whose territory covers a large area of the continent. A total of 69 local breeds and types of dairy, wool, and meat sheep (*Ovis aries*) are maintained here. However, the genetic diversity and maternal origin of these local breeds have not been comprehensively investigated. In this study, we describe the diversity and phylogeny of Russian sheep breeds inhabiting different geographical regions based on the analysis of complete sequences of mitochondrial genomes (mtDNA). Complete mtDNA sequences of the studied sheep were obtained using next-generation sequencing technology (NGS). All investigated geographical groups of sheep were characterized by high haplotype (Hd = 0.9992) and nucleotide diversity (π = 0.00378). Analysis of the AMOVA results showed that genetic diversity was majorly determined by within-population differences (77.87%). We identified 128 haplotypes in all studied sheep. Haplotypes belonged to the following haplogroups: B (64.8%), A (28.9%), C (5.5%), and D (0.8%). Haplogroup B was predominant in the western part of Russia. A high level of mtDNA polymorphism in the studied groups of local sheep indicates the presence of a significant reserve of unique genotypes in Russia, which is to be explored.

## 1. Introduction

Domestic sheep (*O. aries*) is one of the first domesticated, adaptable, and widespread livestock species [1]. The Asian mouflon (*O. orientalis*) is proposed as the ancestor of domestic sheep. Several studies reported that most of the functional genetic diversity in sheep came from mouflon during progressive domestication [2,3].

According to FAO statistics, the population number of domestic sheep worldwide was 1.284 billion in 2021, and 1384 different breeds of sheep have been identified in 2022 [4]. The center of sheep domestication is the Fertile Crescent territory in the Middle East [3]. This was indirectly confirmed by the higher nucleotide diversity of mtDNA sequences of modern sheep in the Levant region [5,6].

The spreading of sheep from the domestication center, mediated by a variety of human activities, took place during the transition from the Mesolithic culture to the Neolithic era (about 8000–9000 years ago) [1]. Chessa et al. [1] used the integration of endogenous retroviruses in the sheep genome as genetic markers and postulated two major waves of sheep migration toward northwest Europe. The first wave began with the spreading of “primitive” sheep from the Fertile Crescent. The second wave of improved sheep migration occurred about 5000 years ago [7].

After approximately 1000 years of domestication, domestic animals and plants expanded and replaced the foraging lifestyle in larger parts of Southwest Asia [8]. Further, they subsequently spread to Europe along two routes: through the Mediterranean Sea and through the Danube Valley. Phylogenetic analysis of lentiviruses of small ruminants supported evidence of these events [9]. The Mediterranean route passed from Cyprus through the Balkan and Apennine peninsulas to Corsica and Sardinia. Later the settlement of Northern Italy and Southern France took place. Then, along the Danube route, sheep spread to Central and Northern Europe [10]. Analyzing mtDNA of Copper Age European sheep, Olivieri et al. [7] concluded that sheep ended up in the Alps a little more than 5000 years ago. In the Iberian Peninsula, the Neolithic settlements were established in Spain (7700–7600 years ago) [11], and then in Portugal (7400–7300 years ago) [5]. However, according to another hypothesis, the introduction of the *Ovis* genus had occurred repeatedly from Africa and Central Europe to the Iberian Peninsula. At the same time, there was a settlement of Northern Europe through the center of domestication through the Caucasus and Russia [12,13].

According to references, the colonization of Asia began from the Middle East to the Mongolian plateau and the Indian subcontinent, from where sheep migrated to the north and southwest of China [14,15]. Sheep distribution across Asia could have happened through several routes as it had happened in Europe [16].

The territory of Russia covering a large part of Eurasia is in the arctic, subarctic, temperate, and partly in the subtropical climatic zones. Thus, the country is characterized by a wide variety of climatic conditions. A total of 69 different sheep breeds have been registered on its territory [17], which produce agricultural products in various climatic conditions. The diversity of environments and long-term adaptation to these resulted in the great genetic diversity of Russian breeds.

The genetic diversity of farm animals plays an important role in the more complete performance of economically significant, fitness-related traits [18]. The conservation of local breeds of livestock is important because local breeds meet human needs in the face of impending climate change scenarios [19] and present a valuable pool of allelic variants for future unknown usage [20]. Moreover, the study of sheep genetics is relevant for understanding aspects of selection, breed diversity, and domestication history as well as for breed sustainable maintenance and conservation. Using genetic markers to address breeding problems in animal husbandry helps to preserve the existing gene pool of the breeds and to improve the economic profitability of the industry.

Mitochondrial DNA (mtDNA) is inherited only through the maternal lineages and has great variability within the same species. In this regard, mtDNA polymorphism analysis is widely used to investigate phylogenetic links and genetic structure of various livestock species [21,22,23,24,25]. Numerous studies including tracing the maternal lineage, and establishing phylogenetic relationships, structure, and diversity of the population have been carried out based on the D-loop mtDNA region and the *CytB* gene [26,27].

Based on the analysis of sheep mtDNA control region polymorphism, seven haplogroups were identified. Haplogroups F and G disappeared, and five others (A, B, C, D, E) are present in modern breeds [5,28]. The first identified haplogroups were A and B, which corresponded to Asian and European types of origin, respectively [28,29]. Later, the third recognized phylogenetic branch, haplogroup C, was discovered [30,31]. Haplogroup C is characterized by a larger genetic diversity than A and B [31]. This haplogroup with low frequency was detected in Portuguese native sheep [32], in individuals from the Caucasus, the Middle East, and Asia. Then, the fourth maternal lineage, haplogroup D, which is the closest to the mouflon (*O. gmelini anatolica Valenciennes*), was identified [12,33,34]. Using D-loop and *CytB* allowed for the detection of the fifth haplogroup E [5]. The latest discovered haplogroups are the rarest ones, which were identified only in animals from Turkey, the Caucasus [5,12,34], Europe [35,36], Tibet [37], and Iran [38]. These haplogroups were formed presumably from 5 to 35 thousand years ago [39].

Thus, a comprehensive study of mtDNA polymorphism allows unlocking the evolution events, which had occurred during the migration of domestic sheep populations throughout the history of their domestication. However, most sheep population studies are based on using the D-loop [40] or the *CytB* gene [41] sequences. The polymorphism of other mitochondrial genes has not been studied so precisely. The study of complete mitochondrial genomes will expand knowledge about the role of mtDNA in the livelihood of sheep.

In this research, for the first time, an investigation of the genetic diversity and structure of Russian sheep populations based on the polymorphism of complete mitochondrial genomes is presented.

## 2. Materials and Methods

### 2.1. Sample Collection

Tissue specimens (ear fragments) of Russian sheep breeds were used as biological material for this research. Sheep samples were provided by the Bioresource Collection of Farm Animals of the L.K. Ernst Federal Science Center for Animal Husbandry (deposited to the collection in the period from 2003 to 2018).

Based on pedigree records, the unrelated individuals were chosen. Additionally, all samples were genotyped using STR markers. The matrix of STR-genotypes was analyzed in ML-Relate software [https://www.montana.edu/kalinowski/software/mL-relate/index.html (accessed on 7 December 2022) [42]. Only unrelated animals were selected for mtDNA sequencing.

The final studied sample included 135 sheep inhabiting six territorial districts of Russia, corresponding to diverse geographical zones (Table 1).

### 2.2. DNA Extraction, PCR Amplification, and Sequencing

DNA was isolated from ear fragments using commercial «DNA-Extran-2» kits (CJSC Syntol, Moscow, Russia) in accordance with the manufacturer’s guidelines. An integrity of DNA was checked by gel electrophoresis in 1% agarose gel via the colorimetric gel documentation system Uvitec FireReader V10 imaging System (Cleaver Scientific Ltd, Rugby, Warwickshire, UK). The OD260/OD280 ratio and concentrations of DNA solutions were measured using a NanoDrop 8000 (Thermo Fisher Scientific, Waltham, MA, USA) and a Qubit 4.0 fluorimeter (Thermo Fisher Scientific, Waltham, MA, USA), respectively. The isolated DNA samples were preserved at −20 °C until further laboratory manipulations.

For the next generation sequencing of complete mitogenomes, six mtDNA fragments with overlapping regions were amplified. Primer sequences for the amplification of target overlapping mtDNA fragments were selected based on the reference sequence of complete mitochondrial genome of sheep, which were presented in the National Center for Biotechnology Information NCBI (GenBank accession number NC_001941.1, accessed on 10 January 2023) according to *O. aries* ARS-UI_Ramb_v2.0 sheep genome version using the online resource Basic Local Alignment Search Tool (BLAST) (https://blast.ncbi.nlm.nih.gov/Blast.cgi (accessed on 10 January 2023)) [43]. The nucleotide sequences of the used primers are shown in Table 2.

The final volume for PCR was 25 µL and contained 10 µL PCR-buffer (2.5½ HF Reaction buffer), 10.25 µL H_2_O, 2.5 µL dNTPs (1 mmol/L), 1 µL primer mix, 0.25 µL SmartTaq HF-FuZZ DNA polymerase (Dialat Ltd., Moscow, Russia), and 1 µL of DNA template.

Quality of PCR amplification was checked by gel electrophoresis in 1% agarose gel using the colorimetric gel documentation system Uvitec FireReader V10 imaging System (Cleaver Scientific Ltd, Rugby, Warwickshire, UK).

PCR products were purified using the «Cleanup Mini kit for the isolation and purification of DNA from agarose gel and reaction mixtures» (JSC Eurogene, Moscow, Russia) and were used to prepare sequencing libraries with NEBNext Ultra II DNA Library Prep Kit for Illumina (New England Biolabs Inc., Ipswich, UK) following the manufacturer’s recommendations. To check the quality, sequencing libraries were submitted to CJSC Syntol (Russia).

Sequencing of the prepared libraries was performed by 300-bp paired-end procedure on MiSeq (Illumina, Inc., San Diego, CA, USA).

### 2.3. Bioinformatics Data Processing

MtDNA sequences were aligned and edited using the MUSCLE algorithm [44] as implemented in MEGA 7.0.26 software (https://www.megasoftware.net/history (accessed on 12 May 2023)) [45].

Measures of genetic diversity parameters including number of polymorphic sites (*S*), average number of nucleotide differences (*K*), number of haplotypes (*H*), haplotype diversity (*Hd*), nucleotide diversity (*π*), and standard error mean (±*SEM*) were calculated in DNASP 6.12.01 software (https://www.bioinformaticshome.com/db/tool/DnaSP (accessed on 7 June 2023)) [46].

Haplogroups for Russian sheep were identified using the MitoToolPy software (http://www.mitotool.org/ (accessed on 15 June 2023)) [47].

The best evolution models were determined using the PartitionFinder 2 program [48] using the Akaike information corrected criterion (AICc) (AICc) [49]. The evolutionary model HKY + I + G was the most optimal.

The phylogenetic relationship was established by the Markov chain Monte Carlo search running with four chains for 10,000,000 generations, with trees being sampled every 500 generations (the first 25% of the trees were discarded as ‘burnin’) in the MrBayes 3.2.7 program (http://nbisweden.github.io/MrBayes/ (accessed on 20 June 2023)) [50] and further visualized in FigTree 1.4.3 (http://tree.bio.ed.ac.uk/software/figtree/ (accessed on 20 June 2023)) [51].

To analyze evolutionary relationships, a median-joining network was constructed in PopART 1.7 software (https://popart.maths.otago.ac.nz/ (accessed on 30 June 2023)) [52].

Wright’s F-statistics [53] and analysis of molecular variance (AMOVA) [54], implemented in Arlequin v 3.5 (http://cmpg.unibe.ch/software/arlequin35/Arl35Downloads.html (accessed on 7 July 2023)) [55], were used to assess population differentiation and to evaluate genetic variability among and within a population.

The demographic history of populations was assessed based on pairwise mismatch distribution analyses using the DnaSP 6.12.01 software [46], and tests for neutrality (Tajima D test, Fu’s Fs test) were performed in DnaSP 6.12.01 [46] and Arlequin v 3.5 [55].

## 3. Results

### 3.1. Genetic Diversity and Haplotype Network

To determine single nucleotide polymorphisms (SNPs) in the complete mitochondrial genomes of sheep, genetic diversity indices were calculated (Table 3).

A total of 804 polymorphic sites were identified in 135 studied animals. The largest number of polymorphic sites (*S* = 468) was determined in sheep of the North Caucasian region, and the smallest number (*S* = 49) was identified in the Volga region. The lowest values of nucleotide diversity and the smallest average number of nucleotide differences were recorded in sheep from the Volga region (*π* = 0.00123 ± 0.00017; *K* = 20.400). Sheep from the Far Eastern region were characterized by the highest values of these indicators (*π* = 0.00589 ± 0.00136; *K* = 97.893).

The results of *Tajima’s D* tests were not significant in all sheep groups which may indicate population stability. However, high negative *Fu’s Fs* values were found in the groups from the Southern and North Caucasian regions. Most likely, this indicates foreign gene flow due to spatial expansion. Perhaps this is due to the presence of representatives of rare haplogroups (C and D), which are characterized by high mtDNA polymorphism, in these geographical regions.

In all studied sheep, a total of 128 haplotypes were identified including 7 haplotypes in FAE, 5 in VOL, 46 in NCN, 17 in SIB, 38 in SOU, and 19 in CEN, respectively. The distribution of haplotypes of complete mitochondrial genomes of sheep is presented in Figure 1 as a median-joining network. There are seven haplotypes, each present in two animals: the haplotype Hap_42 was detected in two samples from the North Caucasian region, Hap_103 and Hap_105 in two samples from the Central, Hap_115 in two samples from Siberian, Hap_89 in two samples from Far Eastern region, Hap_6 in one animal from the North Caucasus and one from the Central region, and Hap_109 in one animal from the Central and one from the Siberian region. The remaining haplotypes were specific to the studied sample. The investigated animals were characterized by high haplotype diversity (*Hd* = 0.9992 ± 0.01). At the same time, the greatest haplotype diversity was determined in sheep from the Southern, Siberian and Volga regions (*Hd* = 1.000 ± 0.006; 1.000 ± 0.020 and 1.000 ± 0.126, respectively).

We checked all identified haplotypes with those published worldwide by comparing nucleotide sequences using the BLAST tool. We did not find the 100% nucleotide similarity (Appendix A). In this regard, we may assume that the haplotypes found in Russian sheep breeds have not been described previously.

### 3.2. Haplogroup Assignment

In the studied groups of sheep four haplogroups were determined (A, B, C, and D), using the MitoToolPy program.

Analysis of median-joining network (Figure 1) and the Bayesian tree (Figure 2) revealed that the most frequent among the studied groups of sheep was haplogroup B, which included 83 haplotypes (64.8%). Haplogroup A was accounted for by 37 haplotypes (28.9%), haplogroup C was represented by seven haplotypes (5.5%), and only one haplotype was assigned to haplogroup D (0.8%). Haplogroups A and B were present in all regional groups of sheep. Haplogroup C was represented by the haplotypes from the North Caucasus and Far Eastern regions, while haplogroup D was found in a single animal from the southern regional group.

### 3.3. Genetic Differentiation between Russian Sheep from Diverse Geographical Regions

To evaluate genetic differentiation of sheep from different regional groups, pairwise *F_ST_* values were calculated (Table 4).

The greatest genetic remoteness was showed between Far Eastern and Central geographical groups (*F_ST_* = 0.43960) and between Volga and Far East geographical groups (*F_ST_* = 0.39243), respectively (Table 4). Obviously, this is due to the territorial remoteness of breeding these groups of sheep. The pairs from the Central and Volga regions (*F_ST_* = 0.00818), as well as the Southern and North Caucasian regions (*F_ST_* = 0.00937) were characterized by the greatest genetic similarity.

Analysis of molecular variance (AMOVA) (Table 5) confirmed a significant genetic differentiation within the studied groups of sheep (77.87%), while differences among studied populations accounted for 22.13% of the genetic variability.

### 3.4. Population Dynamics and Mismatch Distribution of Russian Sheep from Different Regions

The mismatch distribution (distribution of the number of pairwise differences between sequences) for all groups of sheep was multimodal (Figure 3). All studied groups showed at least two peaks, which suggests that several independent events of gene influx from outside occurred in the past. This may be a consequence of the artificial movement of domestic sheep between different populations or of the high genetic diversity within and between haplogroups in originally Neolithic stock.

## 4. Discussion

The study of genetic diversity of sheep provides a better understanding of aspects of breeding and domestication processes. MtDNA is distinguished by its uniqueness and is one of the most informative genetic markers for phylogenetic studies, because mtDNA has a maternal inheritance pattern and allows determining introgression only in females. Here, we investigated complete sequences of mitochondrial genomes of Russian sheep originated from different geographic regions. The phylogeny of mitochondrial genome of the genus *Ovis* is formed from five major haplogroups (A, B, C, D, and E) [6].

Based on analysis of polymorphism of complete mtDNA nucleotide sequences, we identified four sheep haplogroups, including A, B, C, and D, in studied sheep populations.

According to previous studies [6,28,29,56], haplogroup B is typical for sheep of European type of origin. This is confirmed in our study as well, because the most frequent among the studied groups of sheep was haplogroup B, which was represented by 89 animals from all regions. The second largest group (*n* = 38) was haplogroup A, which was also found in all regions, but most of the animals were bred in the Siberian and Far Eastern geographical regions.

As in previous studies [6,57,58], haplogroup A prevailed in Asian sheep populations. Seven animals were assigned to haplogroup C. They were mainly from the North Caucasian region, but two representatives were from the Far Eastern geographical region as well. According to the review by Machová K. et al., the haplogroup C was found in Portugal, the Caucasus, the Middle East, and Asia [59]. In addition, Lv et al. in their study provides a map of haplogroup distribution in the world [14]. According to it, haplogroup C is found mainly in the Middle East, the Caspian Sea region, Northern China, and the Mongolian Plateau. This corresponds to the data obtained in our study. Haplogroups D and E were the rarest [14] as well as in our study only one animal from the Southern ecoregion was belonged to haplogroup D. Taylor et al. [58] previously described a similar distribution of sheep haplogroups. The presence of Haplogroup A and B in all regional groups may indirectly indicate on several ways of settlement from the main center of domestication.

Haplotype (*Hd* = 0.999 ± 0.010) and nucleotide (*π* = 0.00378 ± 0.00027) diversity of Russian sheep were comparable to the values obtained earlier in African sheep (*Hd* = 0.993 ± 0.002, *π* = 0.00254 ± 0.012) [60] and Indian sheep (*Hd* = 0.8502 ± 0.024, *π* = 0.00373 ± 0.012) [57].

The results of the AMOVA determined higher genetic differentiation within the studied groups of sheep (77.87%), rather than among populations (23.13%). Our data were corresponded to the results, which were obtained in Indian sheep (variability among sheep groups is accounted for 11.1%) [57]. Such a pattern of genetic differentiation may indicate on a certain commonality of the maternal origin of the studied groups of sheep from different geographical regions. As already mentioned by several authors e.g., [61], there is hardly any significant breed structure among the maternal lineages of the modern sheep groups, making it difficult to address their relationship to ancient sheep based on mtDNA alone.

*Tajima’s D* neutrality test results were close to zero and statistically insignificant in all sheep populations, suggesting that the evolution of maternal lineages was rather random. However, high negative *Fu’s Fs* (−10.572 **; −8.987 **) values were found in several groups and suggest population expansion. Similar results were obtained in Indian (*Tajima’s D* = −2.1; −0.97; *Fu’s Fs* = −26.12; −25.66) [57] and African sheep (*Tajima’s D* = −0.921; 3.715; *Fu’s Fs* = −24.003; −10.426) [61].

Thus, based on the results, we may summarize that there is a significant reserve of unique genotypes in Russian sheep populations. In future we would continue our investigations of genetic diversity of these populations for better understanding of their developmental history, and genetic relationships.

## 5. Conclusions

On the territory of Russia, there is a large population of domestic sheep with specific genotypes, the genetic potential of which has not been fully explored yet. In the present study, we analyzed the polymorphism of complete mtDNA nucleotide sequences of 135 Russian sheep from different habitats. A large number of haplotypes (*H* = 128) was identified, of which 94.5% (*H* = 121) were unique. High nucleotide and haplotype diversity was revealed that corresponded to high variability of mtDNA. At the same time, the median-joining network of haplotypes formed joined clusters among the populations, which may indicate on a common maternal origin of the studied sheep. The results obtained in this study will be useful for improving strategies for the conservation of sheep genetic resources. Continuing research on large samples will provide a more complete pattern of genetic diversity and demographic history of sheep inhabiting different geographical areas in Eurasia.

## Figures and Tables

**Figure 1 genes-14-01701-f001:**
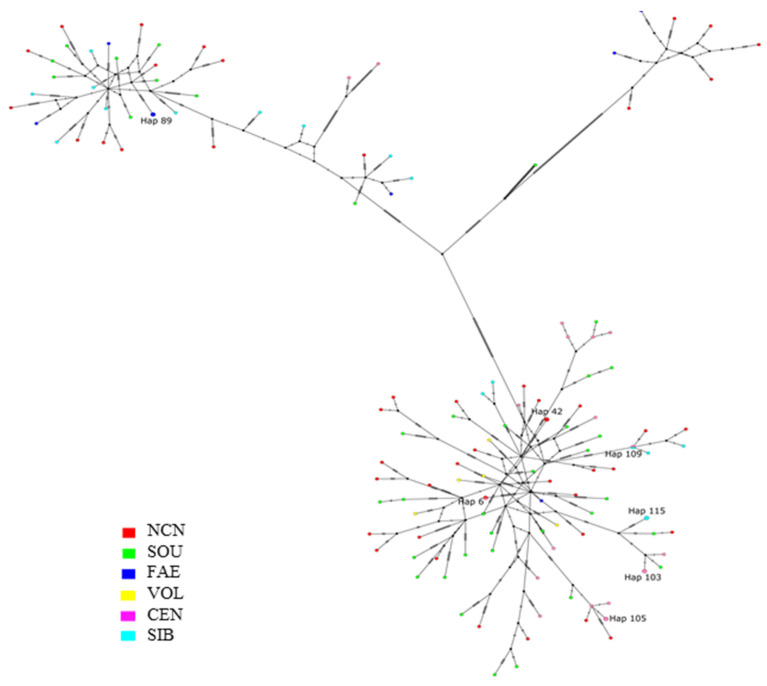
Median-joining network characterizing haplotype connections identified in Russian local sheep breeds based on analysis of the nucleotide sequence of complete mitogenomes. Note: the studied geographical groups of sheep: FAE: Far Eastern; VOL: Volga; NCN: North Caucasian; SIB: Siberian; SOU: Southern; CEN: Central. Colors within each haplogroup ring correspond to different geographical regions. Hap89, Hap42, Hap109, Hap115, Hap 103, and Hap 105 are shared haplotypes.

**Figure 2 genes-14-01701-f002:**
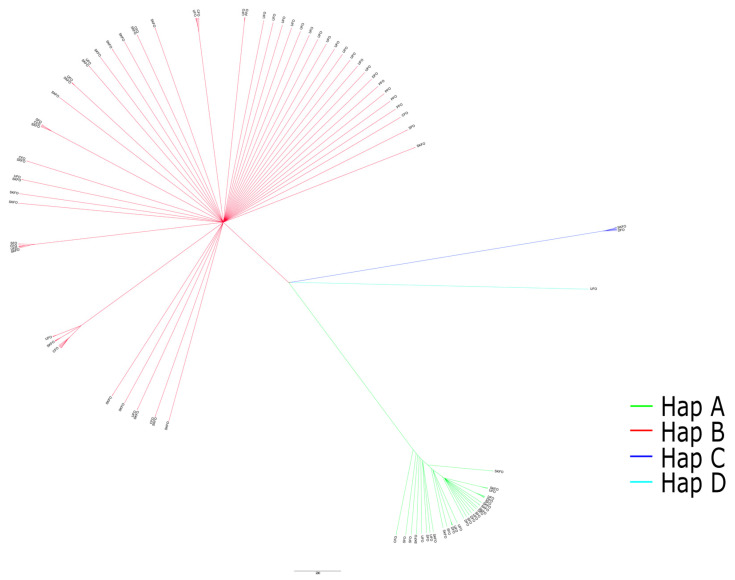
Bayesian phylogenetic tree reflecting genetic connections of Russian sheep breeds based on analysis of nucleotide sequences of complete mitochondrial genomes. Note: the studied geographical groups of sheep: FAE: Far Eastern; VOL: Volga; NCN: North Caucasian; SIB: Siberian; SOU: Southern; CEN: Central. Colors of branches correspond to different haplogroups (A, B, C, and D).

**Figure 3 genes-14-01701-f003:**
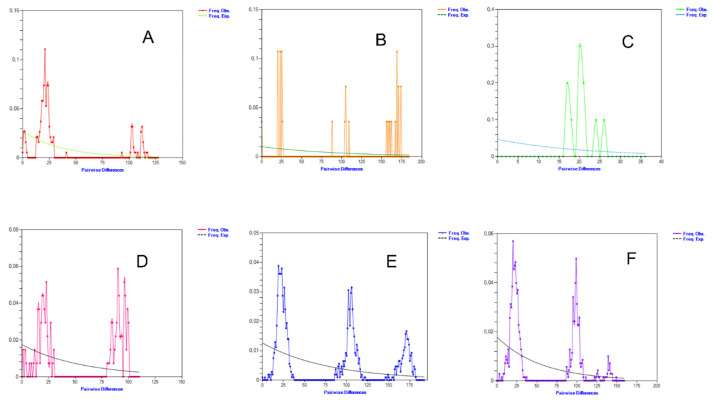
Graph of the mismatch distribution of Russian sheep from different geographical regions: (**A**) Southern; (**B**) North Caucasian; (**C**) Siberian; (**D**) Volga; (**E**) Far Eastern; (**F**) Central. The x and y axes represent the number of pairwise differences and the relative frequency of pairwise comparisons, respectively.

**Table 1 genes-14-01701-t001:** Groups of studied sheep.

Geographical Region	*n* *	Group Composition (Breed)
Far Eastern (FAE)	8	Buubei (*n* = 5)
		Baikal fine-fleeced (*n* = 3)
Volga (VOL)	5	Tsigai (*n* = 5)
North Caucasian (NCN)	47	Dagestan Mountain (*n* = 4)
		Karachaev (*n* = 10)
		Lezgin (*n* = 7)
		Manych Merino (*n* = 5)
		North Caucasian (*n* = 5)
		Soviet Merino (*n* = 3)
		Stavropol (*n* = 5)
		Tushin (*n* = 8)
Siberian (SIB)	17	Altai Mountain (*n* = 5)
		Buryat (*n* = 2)
		Kulundin (*n* = 5)
		Mongol (*n* = 1)
		Tuva short-fat-tailed (*n* = 4)
Southern (SOU)	38	Volgograd (*n* = 8)
		Groznenk (*n* = 7)
		Kalmyk fat-rumped (*n* = 5)
		Karakul (*n* = 6)
		Salsk (*n* = 5)
		Edilbai (*n* = 7)
Central (CEN)	20	Kuibyshev (*n* = 3)
		Kuchugur (*n* = 3)
		Romanov (*n* = 8)
		Russian Longhaired (*n* = 6)

* *n* = sample size.

**Table 2 genes-14-01701-t002:** Primer sequences selected for amplification of sequences of complete mitochondrial genome of sheep (*O. aries*).

Primer	Nucleotide Sequences	Length of Amplification Product, b.p.
mtDNA_For1	5′-AGTACGGCGTAAAGCGTGTT-3′	2987
mtDNA_Rev1	5′-AATGGTGCTCGGTTTGTTTC-3′	
mtDNA_For2	5′-GAAAAGGCCCAAACGTTGTA-3′	3294
mtDNA_Rev2	5′-GATATTATGGCTCATACTATTCCTATATA-3′	
mtDNA_For3	5′-TCCTATATCAACACCTATTCTGATTCTT-3′	3525
mtDNA_Rev3	5′-GGAAGTCAGAATGCGATGGT-3′	
mtDNA_For4	5′-ACACCAAACCCACGCTTATC-3′	3183
mtDNA_Rev4	5′-AAAATTGATTGCTGCGATGGGT-3′	
mtDNA_For5	5′-TGAACGAGTTCACAGCCGAA-3′	2810
mtDNA_Rev5	5′-ATTGTAAGTGGTGGGGTTGG-3′	
mtDNA_For6	5′-AGCAATTCCCATAGCCTCCT-3′	3517
mtDNA_Rev6	5′-GGCTGTTGCGGTAGTACTCT-3′	

**Table 3 genes-14-01701-t003:** Indices of genetic diversity and neutrality of six regional groups of Russian local breeds based on mitogenomes.

Group ^1^	*n* ^2^	S ^3^	K ^4^	H ^5^	Hd ^6^ (±SEM) ^7^	π ^8^ (±SEM)	Tajima’s D ^9^	Fu’s Fs ^10^
FAE	8	246	97,893	7	0.964 ±0.077	0.00589 ±0.00136	0.17425 ns^13^	3.582 ns^13^
VOL	5	49	20,400	5	1.000 ±0.126	0.00123 ±0.00017	−1.00076 ns^13^	0.544 ns^13^
NCN	47	468	78,247	46	0.999 ±0.005	0.00471 ±0.00052	−0.97712 ns^13^	−8.987 *^11^
SIB	17	166	56,191	17	1.000 ±0.020	0.00339 ±0.00026	0.58964 ns^13^	−2.097 **^12^
SOU	38	414	55,092	38	1.000 ±0.006	0.00332 ±0.00041	−1.67111 ns^13^	−10.572 *^11^
CEN	20	214	35,926	19	0.995 ±0.018	0.00216 ±0.00059	−1.68045 ns^13^	−2.62 **^12^
Overall	135	804	62,615	128	0.9992 ±0.010	0.00378 ±0.00027	−1.91342 *^11^	−31.631 *^11^

^1^ Groups of sheep: FAE: Far Eastern; VOL: Volga; NCN: North Caucasian; SIB: Siberian; SOU: Southern; CEN: Central; ^2^ *n* == sample number; ^3^
*S* = variable sites; ^4^
*K* = the average number of nucleotide differences; ^5^
*H* = the number of haplotypes; ^6^
*Hd* = haplotype diversity; ^7^
*SEM* = standard error mean; ^8^
*π* = nucleotide diversity; ^9^
*Tajima’s D* = Tajima’s neutrality test; ^10^
*Fu’s Fs* = neutrality test; *^11^ = indicates statistical significance *p* < 0.1, **^12^ = indicates statistical significance *p* < 0.05; ^13^ns = non-significant.

**Table 4 genes-14-01701-t004:** Pairwise *F_ST_* values calculated for the studied geographical groups of sheep.

Groups	FAE ^1^	VOL ^2^	SIB ^3^	NCN ^4^	SOU ^5^	CEN ^6^
FAE ^1^	0					
VOL ^2^	0.39243	0				
SIB ^3^	0.09707	0.29906	0			
NCN ^4^	0.16099	0.04873	0.05781	0		
SOU ^5^	0.29847	0.03248	0.10804	0.00937	0	
CEN ^6^	0.43960	0.00818	0.28091	0.07288	0.04058	0

^1^ FAE: Far Eastern; ^2^ VOL: Volga; ^3^ NCN: North Caucasian; ^4^ SIB: Siberian; ^5^ SOU: Southern; ^6^ CEN: Central.

**Table 5 genes-14-01701-t005:** AMOVA results for the studied groups of sheep.

Source of Variation	Freedom Degrees d.f.	Sum of Squares, SS	Variance Components, VC	Percentage of Variation V %
Among populations	2	249.322	10.20234	22.13
Within populations	27	969.312	35.90044	77.87
Total	29	1218.633	46.10277	

## Data Availability

The complete nucleotide sequences of mitochondrial genomes of the studied sheep were deposited as population sets according to their geographical origin to NCBI GenBank (accessed on 16 August 2023) under accession numbers OR459640-OR459774.

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
