# Peer review of "Phylogenetic Analysis of Russian Native Sheep Breeds Based on mtDNA Sequences"

_genes, 2023, doi:10.3390/genes14091701_

Round 1
Reviewer 1 Report
Dear Authors,
Congratulations on the idea of phylogenetic analysis of russion local sheep breeds using mitochondrial DNA. Undoubtedly, this would contribute to clarifying the origin of today's local breeds in the vast Russian territory. I accept your article within a phylogenetic analysis, but you put in the title of the article a study of genetic polymorphism in mitochondrial DNA. This title is very engaging in terms of sampling. If you aim to study genetic polymorphism, this obliges you to sample from unrelated animals!!! This is a basic rule of thumb when you studying genetic polymorphisms, regardless of what markers you use or the whole genome. In some breeds, the number of samples taken is too low /3 or 4/, which is extremely insufficient when studying genetic diversity! Therefore, I recommend that you must give an accurate description of how the samples were taken in the herds or how representative they are of the breeds!!!
If you are of the position that in order to establish the genetic polymorphism of mitochondrial DNA it is not necessary for the animals to be unrelated to each other, you must provide the necessary evidence or arguments. Please explain this in the "Introduction" section and give your comments.
If you cannot provide the necessary arguments or that for your research it is not necessary for the animals to be unrelated to each other, then I recommend you change the title of the article...for example, such as: Phylogenetic analysis of Russian native breeds of sheep.....or something similar.....
I have the following recomendation on your article:
1. In the introduction section, the second paragraph, you as authors of the article admitted to quoting inaccurate information about the number of sheep in the world and the number of breeds in the worls. I recommend you to go to the FAO website and take accurate information about the number of sheep in the world (they are 1.284 billion /2021/). Youvshould find more accurate sources of information about the number of sheep breeds in the world!!! They are 1384 breeds/2022/. Please cite reliable sources and accurate information!!!
Otherwise, the laboratory part was completed correctly. The methodology for sample analysis is explained in sufficient detail. 8 softwares are used to discover median-joining network characterizing the haplotype relationships in Russian local breeds based on the analysis of the nucleotide sequence of complete mitogenomes. Congratulations! But, if the samples are not taken correctly then the results are false!
Author Response
Dear Reviewer, we express our sincere gratitude for your time and valuable comments, which helped us to improve our manuscript.
Point 1: Dear Authors,
Congratulations on the idea of phylogenetic analysis of russion local sheep breeds using mitochondrial DNA. Undoubtedly, this would contribute to clarifying the origin of today's local breeds in the vast Russian territory. I accept your article within a phylogenetic analysis, but you put in the title of the article a study of genetic polymorphism in mitochondrial DNA. This title is very engaging in terms of sampling. If you aim to study genetic polymorphism, this obliges you to sample from unrelated animals!!! This is a basic rule of thumb when you studying genetic polymorphisms, regardless of what markers you use or the whole genome. In some breeds, the number of samples taken is too low /3 or 4/, which is extremely insufficient when studying genetic diversity!!! Therefore, I recommend that you must give an accurate description of how the samples were taken in the herds or how representative they are of the breeds
If you are of the position that in order to establish the genetic polymorphism of mitochondrial DNA it is not necessary for the animals to be unrelated to each other, you must provide the necessary evidence or arguments. Please explain this in the "Introduction" section and give your comments.
If you cannot provide the necessary arguments or that for your research it is not necessary for the animals to be unrelated to each other, then I recommend you change the title of the article...for example, such as: Phylogenetic analysis of Russian native breeds of sheep.....or something similar.....
Response 1: Dear Reviewer, we express our sincere gratitude for your time and valuable comments, which helped us to improve our manuscript.
The animals for our study were selected based on pedigree records (if such were available). All animals were genotyped with STR-markers, which were used to check for kinship in ML-Relate software (Kalinowski ST, AP Wagner, ML Taper (2006). ML-Relate: a computer program for maximum likelihood estimation of relatedness and relationship. Molecular Ecology Notes 6:576-579). Thus, we selected only unrelated animals.
We agreed that sample sizes for some breeds are too low. Therefore we attempted to focus on the ecological groups, not separate breeds.
We agreed that the previous title of the manuscript was misleading. In this regard, we changed it as «Phylogenetic analysis of Russian native sheep breeds based on mtDNA sequences».
Point 2: I have the following recomendation on your article:
- In the introduction section, the second paragraph, you as authors of the article admitted to quoting inaccurate information about the number of sheep in the world and the number of breeds in the worls. I recommend you to go to the FAO website and take accurate information about the number of sheep in the world (they are 1.284 billion /2021/). Youvshould find more accurate sources of information about the number of sheep breeds in the world. They are 1384 breeds/2022/.
Please cite reliable sources and accurate information!!!
Response 2: Thank your very much for the comment. We made relevant changes.
Point 3: Otherwise, the laboratory part was completed correctly. The methodology for sample analysis is explained in sufficient detail. 8 softwares are used to discover median-joining network characterizing the haplotype relationships in Russian local breeds based on the analysis of the nucleotide sequence of complete mitogenomes. Congratulations! But, if the samples are not taken correctly then the results are false!
Response 3: We selected sheep samples based on pedigree records and STR-genotypes. We may assume that we took the samples correctly. We added relevant information in 2.1. Sample collection.

Reviewer 2 Report
The present study provides missing information on the genetic diversity of sheep from a vast region of Eurasia. Although the sample size of the studied samples is small -135 sheep, the results show a relatively representative high genetic diversity. The present study provides missing information on the genetic diversity of sheep from a vast region of Eurasia. Although the sample size of the studied samples is small -135 sheep, the results show a relatively representative high genetic diversity.
The authors identified 128 haplotypes in all studied sheep of which 94.5% (H = 121) are unique. Haplotypes were belonged to following haplogroups: B (64.8%), A (28.9%), C (5.5%), and D (0.8%).
I have a few questions and comments about the text.
1. Why is the sequence data not deposited in Genbank? I recommend that they be deposited as a population set.
2. To check all new haplotypes with those published worldwide. This should be reflected as text in the Results.
3. Revise the text. Replace the word - private in all text
>Most haplotypes were private, i.e. present in a single sample. There are seven haplotypes, each present in two animals: the haplotype 206 Hap_42 was detected in two samples from the North Caucasian ecoregion; Hap_103 and 207 Hap_105 in two samples from the Central, Hap_115 in two samples from Siberian, Hap_89 208 in two samples from Far Eastern ecoregion, Hap_6 in one animal from the North Caucasus 209 and one from the Central ecoregion and Hap_109 in one animal from the Central and one 210 from the Siberian region. <
4. The sentence >The remaining haplotypes were private or specific to the population.< is not correct because of small samples.
5. The use of >ecogroups of sheep< in the context of mitochondrial genetic diversity is incomprehensible to me. In the discussion, describe the differences or similarities between the different geographic regions.
6. A comparative analysis of population studies in sheep from neighboring regions in Eurasia is also lacking.
Author Response
Dear Reviewer, we express our sincere gratitude for your time and valuable comments, which helped us to improve our manuscript.
Point 1: The present study provides missing information on the genetic diversity of sheep from a vast region of Eurasia. Although the sample size of the studied samples is small -135 sheep, the results show a relatively representative high genetic diversity. The present study provides missing information on the genetic diversity of sheep from a vast region of Eurasia. Although the sample size of the studied samples is small -135 sheep, the results show a relatively representative high genetic diversity.
The authors identified 128 haplotypes in all studied sheep of which 94.5% (H = 121) are unique. Haplotypes were belonged to following haplogroups: B (64.8%), A (28.9%), C (5.5%), and D (0.8%).
I have a few questions and comments about the text.
- Why is the sequence data not deposited in Genbank? I recommend that they be deposited as a population set.
Response 1: Dear Reviewer, we express our sincere gratitude for your time and valuable comments, which helped us to improve our manuscript.
Following recommendation, we have deposited the sequence as a population set in Genbank under accession numbers OR459640-OR459774 (we obtained the accession numbers; however, sequences have not been published yet).
Point 2: To check all new haplotypes with those published worldwide. This should be reflected as text in the Results.
Response 2: To infer haplotype homology, we checked all identified haplotypes with those published worldwide by comparing of nucleotide sequences by using BLAST tool. We did not find the 100% nucleotide simi-larity (Table S1). In this regard, we may assume that the haplotypes found in Russian sheep breeds have not been described previously.
Point 3. Revise the text. Replace the word - private in all text
>Most haplotypes were private, i.e. present in a single sample. There are seven haplotypes, each present in two animals: the haplotype 206 Hap_42 was detected in two samples from the North Caucasian ecoregion; Hap_103 and 207 Hap_105 in two samples from the Central, Hap_115 in two samples from Siberian, Hap_89 208 in two samples from Far Eastern ecoregion, Hap_6 in one animal from the North Caucasus 209 and one from the Central ecoregion and Hap_109 in one animal from the Central and one 210 from the Siberian region. <
Response 3: We made relevant revisions in the text.
Point 4. The sentence >The remaining haplotypes were private or specific to the population.< is not correct because of small samples.
Response 4: We corrected the sentence.
Point 5. The use of >ecogroups of sheep< in the context of mitochondrial genetic diversity is incomprehensible to me. In the discussion, describe the differences or similarities between the different geographic regions.
Response 5: We made relevant corrections.
Point 6. A comparative analysis of population studies in sheep from neighboring regions in Eurasia is also lacking.
Response 6: Thank you veru much for the suggestion! We will focus on this analysis in our future study.

Reviewer 3 Report
This is an interesting and well-written manuscript aimed to study genetic diversity and phylogeny of Russian sheep breeds through the analysis of complete mtDNA. Material and methods are very interesting and results are clear and concise; however, a more in deep biological discussion of the research findings is recommended. Also, I suggest to consider next minor comments:
- Lines 26-27: Replace “The predominance of Haplotype B” by “Haplotype B was the predominant”.
- Line 58: replace “Olivieri C. et al.” by “Olivieri et al.”.
- Line 94: Replace “[30; 31]” by “[30, 31]”.
- Line 101: Replace “[5; 12; 34]” by “[5, 12, 34]”.
- Line 101: Replace “[35; 36]” by “[35, 36]”.
- Line 125: Please provide the name of the DNA extraction kit or a brief description of the DNA extraction procedure.
- Line 136: Provide the accession date to the GenBank.
- Line 137: Provide the website for the bioinformatics resource.
- Line 140: What was the version of the sheep genome?
- Lines 157, 161, 162, 169, 171, 173 and 176: Provide the website for the bioinformatics resources.
- Lines 214 and 215: Separate numbers from signs within the round bracket.
- Line 269: I suggest improving the Discussion section by adding more biological explanation of the findings according to the research hypothesis.
- Line 290; Replace “Lv, F.H. et al.” by “Lv et al.”.
- Line 295: Replace “Taylor T. et al.” by “Taylor et al.”.
- Line 303: Remove the word “analysis” as the AMOVA abbreviation already includes this word.
- Line 361, 397 and 500: Remove the continuous dot signs.
- Line 358: In References section all Journal names should be abbreviated.
- Lines 379, 385, 397, 447, 457, 464, 466, 470, 477 and 506: Only the first letter of the article title should be capitalized.
Author Response
Dear Reviewer, we express our sincere gratitude for your time and valuable comments, which helped us to improve our manuscript.
Point 1: This is an interesting and well-written manuscript aimed to study genetic diversity and phylogeny of Russian sheep breeds through the analysis of complete mtDNA. Material and methods are very interesting and results are clear and concise; however, a more in deep biological discussion of the research findings is recommended. Also, I suggest to consider next minor comments:
Lines 26-27: Replace “The predominance of Haplotype B” by “Haplotype B was the predominant”.
Response 1: Dear Reviewer, we express our sincere gratitude for your time and valuable comments, which helped us to improve our manuscript.
Replaced
Point 2: Line 58: replace “Olivieri C. et al.” by “Olivieri et al.”.
Response 2: Corrected
Point 3: Line 94: Replace “[30; 31]” by “[30, 31]”.
Response 3: Corrected
Point 4: Line 101: Replace “[5; 12; 34]” by “[5, 12, 34]”.
Response 4: Corrected
Point 5: Line 101: Replace “[35; 36]” by “[35, 36]”.
Response 5: Corrected
Point 6: Line 125: Please provide the name of the DNA extraction kit or a brief description of the DNA extraction procedure.
Response 6: The name of DNA extraction kit is «DNA-Extran-2» (CJSC Syntol, Russia). It is used for DNA extraction from tissues (ears fragments, muscles, etc.). The kit includes Solution 1, Solution 2, Solution 3, merkaptoethanol, Proteinase K and Washing Buffer.
Point 7: Line 136: Provide the accession date to the GenBank.
Response 7: We provided the accession date.
Point 8: Line 137: Provide the website for the bioinformatics resource.
Response 8: We provided the website.
Point 9: Line 140: What was the version of the sheep genome?
Response 9: It was used Ovis aries ARS-UI_Ramb_v2.0 genome version.
Point 10: Lines 157, 161, 162, 169, 171, 173 and 176: Provide the website for the bioinformatics resources.
Response 10: We provided relevant websites.
Point 11: Lines 214 and 215: Separate numbers from signs within the round bracket.
Response 11: Corrected
Point 12: Line 269: I suggest improving the Discussion section by adding more biological explanation of the findings according to the research hypothesis.
Response 12: Thank you very much for suggection. We will wrire more deeper Discussion in our future research.
Point 13: Line 290; Replace “Lv, F.H. et al.” by “Lv et al.”.
Response 13: Corrected
Point 14: Line 295: Replace “Taylor T. et al.” by “Taylor et al.”.
Response 14: Corrected
Point 15: Line 303: Remove the word “analysis” as the AMOVA abbreviation already includes this word.
Response 15: Removed
Point 16: Line 361, 397 and 500: Remove the continuous dot signs.
Response 16: Removed
Point 17: Line 358: In References section all Journal names should be abbreviated
Response 17:. Revised.
Point 18: Lines 379, 385, 397, 447, 457, 464, 466, 470, 477 and 506: Only the first letter of the article title should be capitalized.
Response 18: Corrected

Round 2
Reviewer 1 Report
Dear authors, I am pleased that you have taken into account my notes and recommendations. I would like to assure you that they were done really with the aim of improving the manuscript and especially the methodological part. The Material and Methods section must always describe truthfully, accurately and clearly how the samples were taken and how the research was conducted. Now your manuscript looks much better and is acceptable for printing.